# Spatial cluster analysis of *Plasmodium vivax* and *P. malariae* exposure using serological data among Haitian school children sampled between 2014 and 2016

**Adan Oviedo**[1], **Camelia Herman**[2], **Alaine Knipes**[2], **Caitlin M. Worrell**[2], **LeAnne M. Fox**[2], **Luccene Desir**[3], **Carl Fayette**[4], **Alain Javel**[4], **Franck Monestime**[4], **Kimberly E. Mace**[2], **Michelle A. Chang**[2], **Jean F. Lemoine**[5], **Kimberly Won**[2], **Venkatachalam Udhayakumar**[2], **Eric Rogier**[2]*

**1** Rollins School of Public Health, Emory University, Atlanta, Georgia, United States of America, **2** Division of Parasitic Diseases and Malaria, Centers for Disease Control and Prevention, Atlanta, Georgia, United States of America, **3** The Carter Center, Atlanta, Georgia, United States of America, **4** IMA World Health, Port-au-Prince, Haiti, Hawaii, United States of America, **5** Programme National de Contrôle de la Malaria, Ministère de la Santé Publique et de la Population (MSPP), Port-au-Prince, Haiti, Hawaii, United States of America

* erogier@cdc.gov

**Data Availability Statement:** All relevant data are within the manuscript and its Supporting Information files.

## Abstract

### Background

Estimation of malaria prevalence in very low transmission settings is difficult by even the most advanced diagnostic tests. Antibodies against malaria antigens provide an indicator of active or past exposure to these parasites. The prominent malaria species within Haiti is *Plasmodium falciparum*, but *P. vivax* and *P. malariae* infections are also known to be endemic.

### Methodology/Principal findings

From 2014–2016, 28,681 Haitian children were enrolled in school-based serosurveys and were asked to provide a blood sample for detection of antibodies against multiple infectious diseases. IgG against the *P. falciparum*, *P. vivax*, and *P. malariae* merozoite surface protein 19kD subunit ($MSP1_{19}$) antigens was detected by a multiplex bead assay (MBA). A subset of samples was also tested for *Plasmodium* DNA by PCR assays, and for *Plasmodium* antigens by a multiplex antigen detection assay. Geospatial clustering of high seroprevalence areas for *P. vivax* and *P. malariae* antigens was assessed by both Ripley's K-function and Kulldorff's spatial scan statistic. Of 21,719 children enrolled in 680 schools in Haiti who provided samples to assay for IgG against $PmMSP1_{19}$, 278 (1.27%) were seropositive. Of 24,559 children enrolled in 788 schools providing samples for $PvMSP1_{19}$ serology, 113 (0.46%) were seropositive. Two significant clusters of seropositivity were identified throughout the country for *P. malariae* exposure, and two identified for *P. vivax*. No samples were found to be positive for *Plasmodium* DNA or antigens.

**Funding:** The author(s) received no specific funding for this work.

**Competing interests:** The authors have declared that no competing interests exist.

## Conclusions/Significance

From school-based surveys conducted from 2014 to 2016, very few Haitian children had evidence of exposure to *P. vivax* or *P. malariae*, with no children testing positive for active infection. Spatial scan statistics identified non-overlapping areas of the country with higher seroprevalence for these two malarias. Serological data provides useful information of exposure to very low endemic malaria species in a population that is unlikely to present to clinics with symptomatic infections.

## Author summary

*P. falciparum* is the dominant malaria species worldwide and is often the primary, or only, focus of malaria surveys. For this reason, other human malarias (*P. vivax*, *P. malariae*, *P. ovale*) may be co-endemic in the same population but left unobserved by non-microscopy strategies as countries do not invest in diagnostic capacity to detect these species. Additionally, as these non-falciparum malarias may circulate subpatently, epidemiological measurements through health facility reporting do poorly at estimating the true burden in the population. Antibodies against malaria antigens may exist in persons long after malaria parasites have been cleared and offer an indicator of malaria exposure rather than a test for active infection. For areas in the world with multiple co-endemic malaria species, testing for antibodies against species-specific antigens can allow evaluation of the population burden of all human malarias, not just the dominant or most clinically-relevant species. Serological data can further assist countries as they work towards elimination of all malarias within their borders.

## Introduction

Identification of Plasmodium reservoirs in low transmission settings relies on conventional diagnostics (microscopy and RDT) which underestimate the true incidence and prevalence [1,2]. When compared to microscopy and RDTs, antibody-based assays are a more powerful method for detection of active infection or past exposure to a *Plasmodium* spp. by expanding the time window for diagnosis. Although there is generally an overlap between active infection and exposure data [3,4], attaining exposure data through serology explores malaria transmission by approximating past malaria infections. Antibodies produced by activated B-cells have been identified for all four stages of the human malaria lifecycle (sporozoite, liver stage, blood stage, and sexual stage) [5–7], though the highest titers are elicited to blood-stage antigens [8]. IgG antibodies against the merozoite surface protein 1 (MSP1) antigens produced by different human malarias have been modeled to show half-lives in systemic circulation for years to decades, though this has been studied much more extensively for PfMSP1 compared to the other non-falciparum isoforms [9–11]. Recent advances in assay development have allowed serological data to be collected simultaneously in a multiplex format, making data collection for multiple *Plasmodium* spp. more feasible [5].

As a result of efforts in mass-microscopy to assist malaria elimination efforts in Haiti in the 1960s, three species of *Plasmodia* were identified in the country (*P. falciparum*, *P. vivax*, *P. malariae*), with *P. falciparum* single-species infections accounting for approximately 97% of all persons with parasitemia [12–14]. Though great success was seen in reducing parasite

prevalence in the population through intensive vector control and mass-drug administration campaigns, the elimination program was suspended in the late 1960s with all three species still found throughout the country [13]. Since that time, only a handful of published reports and studies have documented the presence of *P. vivax* or *P. malariae* in Haiti [15–17]. Additionally, after the 2010 earthquake in Haiti, the large-scale deployment of histidine-rich protein 2 (HRP2)-based rapid diagnostic tests (RDTs) led to further under-diagnosis of *P. vivax* and *P. malariae* since these tests only identify *P. falciparum* infections, unless microscopy is also being utilized [18,19]. We present here malaria serological data from school-based surveys from 2014–2016 in Haiti. A multiplex bead assay (MBA) was used to collect IgG data from samples collected in these surveys to assess the seroprevalence of anti-*Plasmodium* MSP1 antibodies with specific focus on PvMSP1 and PmMSP1 responses and exploratory spatial data investigation of potential seroprevalence clustering in Haiti. This study aims to utilize IgG data to determine spatial transmission patterns for the very low endemic malaria species of *P. malariae* and *P. vivax* in Haiti.

## Methods

### Human subjects

Samples were collected from 2014 to 2016 as part of lymphatic filariasis (LF) transmission assessment surveys (TASs) in Haiti, with integration of malaria RDTs and microscopy for soil-transmitted helminths in stool specimens [20]. Fingerprick blood was collected on filter papers (TropBio filter wheels, Cellabs, Sydney, Australia), dried (creating a dried blood spot, DBS), and packaged individually with desiccant for later laboratory analysis at the Centers for Disease Control and Prevention in Atlanta, GA. This activity was considered a program evaluation activity by CDC Human Subjects Office (#2014–256). Persons consented to future laboratory testing of DBS, and CDC laboratory staff did not have access to any personal identifiers.

### Survey design

Surveys were conducted in evaluation units (EUs) that had met World Health Organization (WHO) criteria to conduct LF TAS, with the current WHO recommendation to conduct a school-based TAS in areas where the net primary-school enrollment rate is ≥75%. Haitian school enrollment data for 2014 was utilized along with population census data to determine sampling approach employed in each EU, which are program defined and dependent on baseline LF prevalence found during initial mapping surveys [21]. Using the tables provided by WHO [22], the target sample size and the critical cut-off threshold was determined, and if the number of positive children identified fell below the critical cut-off threshold, the recommendation was made to stop mass-drug administration (MDA) for LF. Upon completion of MDA, TAS surveys were conducted to determine whether populations have reached the critical threshold of infection prevalence (<2% antigenemia), below which LF transmission is likely no longer sustainable [20]. At each school, all children in second grade (overwhelmingly aged 6 or 7 years, Tables 1 and 2) were asked to participate.

### Laboratory assays

**Elution of blood from DBS.** For the antibody and antigen detection assays, whole blood was eluted from a single tab of the TropBio filter wheels to provide sample for assay. A single DBS tab (10 μL whole blood) was rehydrated in a blocking buffer (PBS pH 7.2, 0.5% Polyvinyl alcohol (SigmaAldrich, St. Louis, MO) 0.5% polyvinylpyrrolidine (SigmaAldrich), 0.1% casein

**Table 1. Summary of *Plasmodium malariae* serology samples collected in Haiti between 2014–2016.**

| Variable | | Total | Seropositive |
|---|---|---|---|
| | | N | n (%) |
| Schools [a] | | 680 | 189 (27.8) |
| Participants | | 21,719 | 278 (1.3) |
| Age | | | |
| | 6 | 9,015 | 124 (1.4) |
| | 7 | 12,184 | 154 (1.3) |
| | 8–10 | 4 | 0 (0.0) |
| Gender | | | |
| | Female | 10,531 | 143 (1.4) |
| | Male | 10,674 | 135 (1.3) |
| Department | | | |
| | Centre | 1,213 | 46 (3.8) |
| | Grand'Anse | 1,666 | 12 (0.7) |
| | L'Artibonite | 3,189 | 15 (0.5) |
| | Nippes | 44 | 0 (0.0) |
| | Nord | 9,409 | 116 (1.2) |
| | Nord-Est | 1,625 | 31 (1.9) |
| | Nord-Ouest | 2,368 | 35 (1.5) |
| | Ouest | 15 | 0 (0.0) |
| | Sud | 2,190 | 23 (1.1) |
| | Sud-Est | - | - |

[a] One or more students determined seropositive categorizes a school as seropositive

(ThermoFisher, Waltham, MA), 0.5% bovine serum albumin (SigmaAldrich), 0.3% Tween-20, 0.05% sodium azide, and 0.01% *E. coli* extract to prevent non-specific binding) to a final dilution of 1:20. Eluted blood samples were stored at 4°C until assayed.

**Bead-based IgG detection.** Three separate bead regions (Bio-Plex non-magnetic beads, BioRad) were coupled with malaria antigens for IgG capture and subsequent detection. The antigens in the multiplex panel have all been reported before [5,23], and were the *Plasmodium falciparum* merozoite surface protein 1 19kD fragment (PfMSP1$_{19}$; coupled at pH 5 at 20 µg/mL), *P. vivax* merozoite surface protein 1 19kD fragment (PvMSP1$_{19}$; coupled at pH 5 at 20 µg/mL), and *P. malariae* merozoite surface protein 1 19kD fragment (PmMSP1$_{19}$; coupled at pH 5 at 20 µg/mL). Overall sequence consensus among these three recombinant 19kD antigens was low at 34.4%.

All IgG assay reagents were diluted in buffer containing PBS, 0.05% Tween20, 0.5% bovine serum albumin (SigmaAldrich), and 0.02% NaN$_3$. Hyperimmune positive and negative controls were included on each IgG detection plate to ensure appropriate assay performance. A bead mix was prepared so approximately 1,000 beads/region would be in each assay well. Samples (50 µL of 1:200 dilution whole blood) were incubated with beads for 90 min at room temperature (25°C) under gentle shaking protected from light in MultiScreen-BV filter plates (SigmaAldrich). After three washes (wash buffer: PBS, 0.05% Tween 20), beads were incubated with 50 µL biotinylated detection antibody (a mixture of 1:500 anti-hIgG and 1:625 anti-hIgG$_4$, both produced by Southern Biotech, Birmingham, AL) for 45 min with same incubation conditions as above. After three washes, 50 µL streptavidin-phycoerythrin (Invitrogen, Waltham, MA) was added to all wells (1:250x of 1 mg/mL) for a 30 min incubation. After three washes, samples beads were incubated with 50 µL reagent buffer for 30 min, washed once, and

**Table 2. Summary *Plasmodium vivax* serology samples collected in Haiti between 2014–2016.**

| Variable | | Total | Seropositive |
|---|---|---|---|
| | | N | n (%) |
| Participants | | 24,559 | 113 (0.5) |
| Schools [a] | | 788 | 93 (11.8) |
| Age | | | |
| | 6 | 10,152 | 46 (0.45) |
| | 7 | 13,136 | 54 (0.41) |
| | 8–10 | 741 | 10 (1.35) |
| Gender | | | |
| | Female | 12,045 | 54 (0.4) |
| | Male | 11,986 | 56 (0.5) |
| Department | | | |
| | Centre | 1,213 | 5 (0.4) |
| | Grand'Anse | 1,666 | 2 (0.1) |
| | L'Artibonite | 3,189 | 0 (0.0) |
| | Nippes | 1,366 | 14 (1.0) |
| | Nord | 9,184 | 43 (0.5) |
| | Nord-Est | 3,115 | 37 (1.2) |
| | Nord-Ouest | 2,368 | 6 (0.3) |
| | Ouest | 71 | 0 (0.0) |
| | Sud | 2,234 | 1 (0.0) |
| | Sud-Est | 153 | 5 (3.3) |

[a] One or more students determined seropositive categorizes a school as seropositive

resuspended in 100 μL PBS. Assay plates were briefly shaken and read on a Bio-Plex 200 machine (BioRad) by generating the median fluorescence intensity (MFI) for 50 beads. The final measure, denoted as MFI minus background (MFI-bg), was reported by subtracting MFI values of beads on each plate only exposed to sample diluent during the sample incubation step. The MFI-bg threshold for true positive IgG assay signal against *Plasmodium* antigens was ascertained if the sample MFI-bg was higher than the mean + 3SD of the MFI-bg signal of a panel of 92 known anti-malaria IgG negative DBS samples.

The testing of TAS samples was already underway before the $PmMSP1_{19}$ antigen was available for use. For this reason, the first two EUs consisting of the Sud-Est and Nippes departments in southern Haiti are lacking $PmMSP1_{19}$ IgG data.

**Sample selection for detection of active Plasmodium infections.** Persons (and animals) with active *P. malariae* or *P. vivax* infections typically show high IgG titers to the $MSP1_{19}$ antigens produced by those malarias [23–27]. These same studies have also shown high specificity among *Plasmodium* $MSP1_{19}$ isoforms, meaning infection with one malaria species will typically produce IgG with homologous binding to only that $MSP1_{19}$ isoform. With these two factors considered, samples with the highest IgG assay signals to the $PmMSP1_{19}$ or $PvMSP1_{19}$ antigens were selected for malaria antigen and PCR assays in an attempt to identify active infections to these malarias. For these additional lab assays, initial target sample selection was approximately 25 of the highest IgG signals for each antigen, and if active infections were found, the next 25 highest responders would also be selected for antigen detection and PCRs.

**Bead-based malaria antigen detection.** For detection of *Plasmodium* antigens, three separate bead regions (Bio-Plex non-magnetic beads, BioRad) were coupled by the EDC/Sulfo-NHS intermediate reaction with antibodies against antigen targets: pan-*Plasmodium* aldolase

(12.5μg/12.5x10$^6$ beads, rabbit IgG anti-aldolase, Abcam, Cambridge, UK), histidine-rich protein 2 (HRP2; 20μg/12.5x10$^6$ beads, mouse anti-*P. falciparum*, Abcam), pan-*Plasmodium* lactate dehydrogenase (pLDH; 12.5ug/12.5x10$^6$ beads, mouse IgG anti-LDH, BBI Solutions, Cardiff, UK). All detection antibodies were previously biotinylated by Thermo Scientific EZ-Link Micro Sulfo-NHS-Biotinylation Kit (ThermoFisher Scientific) according to the manufacturer's protocol.

All antigen detection reagents were diluted in buffer containing PBS, 0.05% Tween20, 0.5% bovine serum albumin (SigmaAldrich), and 0.02% NaN$_3$. Recombinant antigen positive and negative blood controls were included on each antigen detection plate to ensure appropriate assay performance. A bead mix was prepared so approximately 800 beads/region would be in each assay well. Bead mix was pipetted into MultiScreen-BV plates, washed twice with wash buffer, and incubated for 90min with 50μL sample (at 1:20 dilution). After three washes, beads were incubated for 45min with a 50uL mix of detection antibodies (1:1000 anti-aldolase [Abcam], 1:500x of 2:1:1 mixture [BBI Solutions BM355-P4A2:BioRad Pv-pLDH HCA156: BioRad Pf-pLDH HCA158]), and 1:500 anti-HRP2 [Abcam]. Following three washes, 50 μL streptavidin-phycoerythrin (Invitrogen,) was added to all wells (1:250x of 1 mg/mL) for a 30 min incubation. After three washes, sample beads were incubated with 50 μL reagent buffer for 30 min, washed once, and resuspended in 100 μL PBS. Assay plates were shaken briefly and read on a Bio-Plex 200 machine (Bio-Rad) by generating the median fluorescence intensity (MFI) for 50 beads. The final measure, denoted as MFI-bg, was reported by subtracting MFI values from beads on each plate only exposed to sample diluent during the sample incubation step. The MFI-bg threshold for a true positive assay signal was the mean + 3SD of the MFI-bg signal of a panel of known malaria antigen negative DBS samples.

**DNA extraction and Photoelectron-induced Electron Transfer (PET) PCR.** DNA was extracted from DBS by using the QIAamp DNA Mini Kit (Qiagen, Hilden, Germany) as recommended by the manufacturer. Briefly, a DBS tab (equivalent to 10 μL whole blood) was placed into a 1.5 mL tube and processed according to instructions. The DNA was eluted in 150 μL of elution buffer and stored at -20˚C until use. Real-time PCR reactions for *Plasmodium* DNA were carried out as described previously using the multiplex PET-PCR assay as previously described [28], with positive and negative controls included on each PCR plate.

## Exploratory spatial analysis

All data cleaning were performed using SAS (Version 9.4). Geocoordinates were available for all schools enrolled in the TAS. Administrative boundaries for Haitian departments were obtained from geoBoundries (https://www.geoboundaries.org/), a service produced by the William & Mary Geolab.

Haiti is a country with total surface area of 27,750 km$^2$. School latitude and longitude coordinates were geocoded and converted from World Geodetic System 1984 into Universal Transverse Mercator Zone 18 projections using ArcMap (Version 10.5.1). Point pattern spatial randomness measures were calculated using Ripley's K-function through ArcMap. Ripley's K-function quantifies how often events are found within a certain distance of one another and the level of aggregation, randomness, or dispersion within the boundaries of a set of observations [29–31]. If observations are not spatially random, it would suggest evidence of biological, environmental, or sampling association with the clustering of cases. K-function permutations allow for confidence interval calculation around an expected random distribution [31]. K-functions in this study was calculated using a maximum radius of 50 km with 2.5 km intervals. Weighted K-function was calculated using seropositive counts per school with the same parameters. Confidence intervals were calculated using 999 permutations for weighted and

unweighted K-functions. Autocorrelation was calculated using Moran's I in ArcMap. Autocorrelation measures such as Moran's I allow for the quantification of the degree to which values spatially correlate to similar values [31]. If there is no spatial autocorrelation, Moran's I is assumed to equal zero, whereas values above zero indicate autocorrelation of data values among the locations. Moran's I was calculated using inverse distance weights, 999 permutations, and a threshold distance of 20 km. Cluster analysis was performed by Kulldorff's spatial scan statistic using Satscan (Version 9.6) to identify clusters with an increased relative risk of malaria exposure. Kulldorff's scan statistic considers moving "windows" with variable radii ranging from the smallest observed distance to a pre-determined upper bound [31], and was calculated using elliptical windows with a maximum of 50% of the population, using discrete purely spatial Poisson modeling, 999 Monte Carlo simulations, non-overlapping windows, and maximum likelihood estimations with an alpha level of 0.05. In addition, spatial analyses were run after the study area was also divided into northern and southern regions. This was performed in order to compare global results with more localized results and explore the possibility of a biased clustering effect due to the large area of unsampled departments previously mentioned for each species. The north region included the departments Nord, Nord-Est, Nord-Ouest, Artibonite, and Centre which provided a cumulative surface area of 14,315 km$^2$. The southern region included the departments, Grand'Anse, Sud, Nippes, and Sud-Est which provided a cumulative surface area of 7,868 km$^2$. The department of Ouest was excluded for clustering analyses as so few children were enrolled for collection of both *P. malariae* and *P. vivax* serology data.

## Results

### Independence of the anti-*Plasmodium* MSP1$_{19}$ IgG assay signals

Histograms for the log-transformed IgG assay signal to PfMSP1$_{19}$, PvMSP1$_{19}$, and PmMSP1$_{19}$ are shown in Fig 1A. In this low malaria transmission setting, a strong unimodal distribution can be observed which would be the "background" signal of the assay for seronegative individuals' blood samples. The seropositivity thresholds are indicated for each antigen, and seropositive sample signals indicated on the right side of each plot. When comparing the assay signal among the three *Plasmodium* MSP1$_{19}$ species' isoforms, very few children provided a blood sample which had a substantial signal to more than one of the MSP$_{19}$ antigens (Fig 1B).

### Characteristics of children seropositive for *P. malariae* and/or *P. vivax* antigens

A total of 21,719 children were tested for *P. malariae* MSP1 exposure in 680 schools (Table 1), and 24,559 children were tested for *P. vivax* MSP1 exposure in 788 schools (Table 2). Of those children tested, 278 (1.27%) were seropositive for PmMSP1$_{19}$ and 113 (0.46%) seropositive for PvMSP1$_{19}$. Of the 21,415 students with assay results for both species, 12 (0.06%) were seropositive for both the *P. malariae* and *P. vivax* antigens. Of the schools surveyed, 189 (27.8%) enrolled one or more seropositive children for PmMSP1$_{19}$, and 93 (11.8%) schools enrolled one or more seropositive children for PvMSP1$_{19}$. By department (administrative district), *P. malariae* seroprevalence remained relatively similar to national seroprevalence in the Nord (1.2%), Nord-Ouest (1.5%), and Sud (1.1%) departments, but was more variable in other regions (Table 1). For departments with more than one-hundred children sampled, *P. malariae* seroprevalence was highest in Centre (3.8%) and lowest in Artibonite (0.5%). By department, *P. vivax* seroprevalence remained similar to national seroprevalence in the Centre (0.4%), Nord (0.5%), and Nord-Ouest (0.3%) but was more variable in other regions (Table 2).

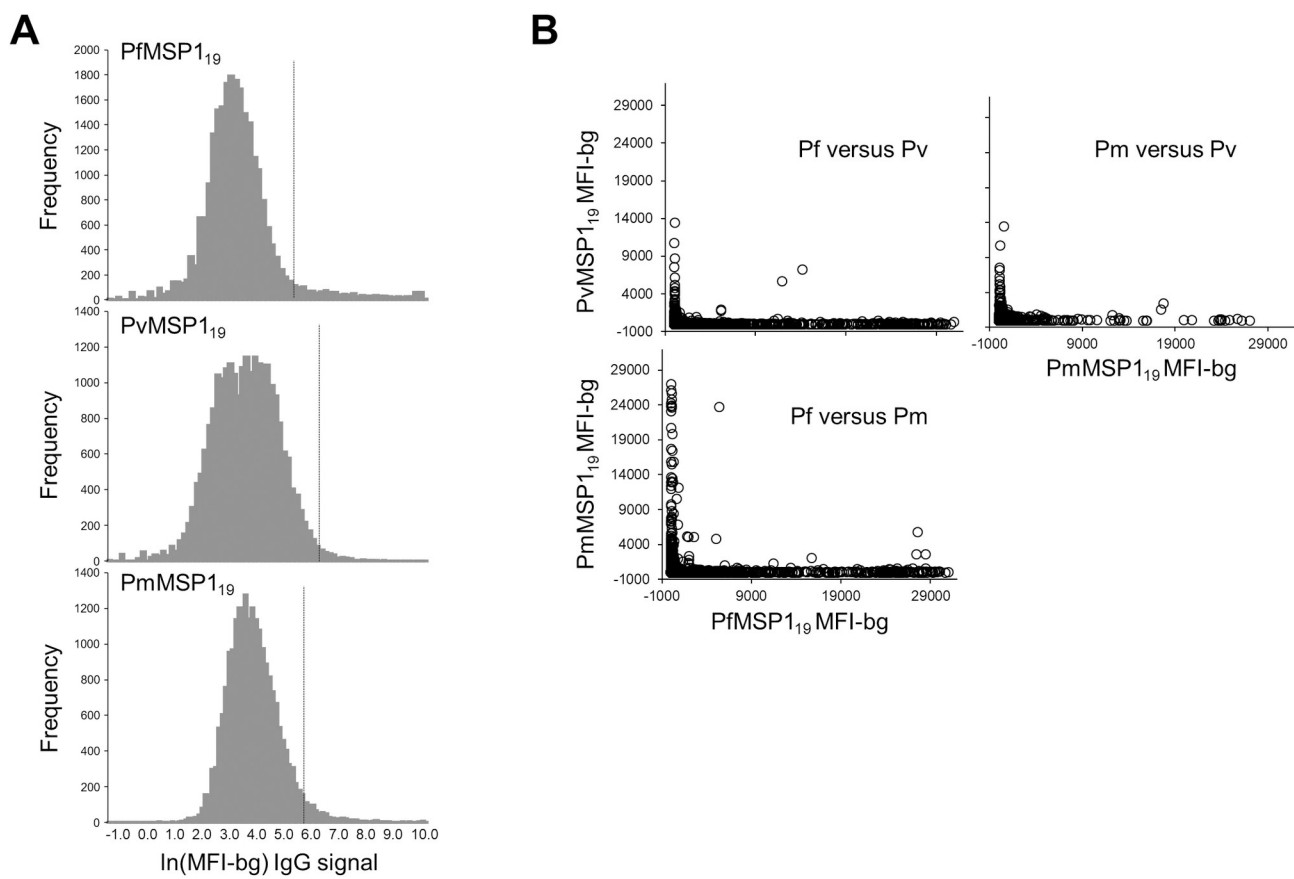

**Fig 1. Distribution and independence of assay signals for IgG against PfMSP1₁₉, PvMSP1₁₉, and PmMSP1₁₉.** A) Histograms of log-transformed assay signal for the three *Plasmodium* antigens. Vertical hashed lines indicate seropositivity threshold for each antigen assay signal. B) Scatterplots of non-transformed assay signal as compared among the three species' MSP1₁₉ isoforms.

For departments with more than one-hundred children sampled, *P. vivax* seroprevalence was highest in Sud-Est (3.3%) and lowest in Artibonite (0.0%). The 12 participants seropositive for both species were widely dispersed across the departments of Sud, Centre, Nord, Nord-Ouest, and Nord-Est. Average age among seropositive children was 6.7 years for *P. vivax* and 6.6 years for *P. malariae*. In addition, among *P. malariae* and *P. vivax* seropositive children, 51.4% and 49.1% were female respectively.

## Spatial distribution of enrolled schools and seropositive children

Spatial distribution of schools enrolled in the 2014–2016 TASs and had *Plasmodium* IgG data collected is shown Fig 2. Indicators on the maps are provided for count of seropositive children per school for *P. malariae* and *P. vivax* antigens.

## Assessment of spatial randomness

Ripley's K function for spatial randomness of schools determined a significant spatial clustering at all intervals tested up to 50 km for both species in northern and southern Haiti (S1A, S1C, S1E and S1G Fig). Weighted K-function of schools tested for *P. malariae* in northern Haiti using seropositive counts as weights suggests a significant spatial aggregation of cases per school at all intervals between 2.5 km and 15 km with a peak at 12.5 km (S1B Fig). The

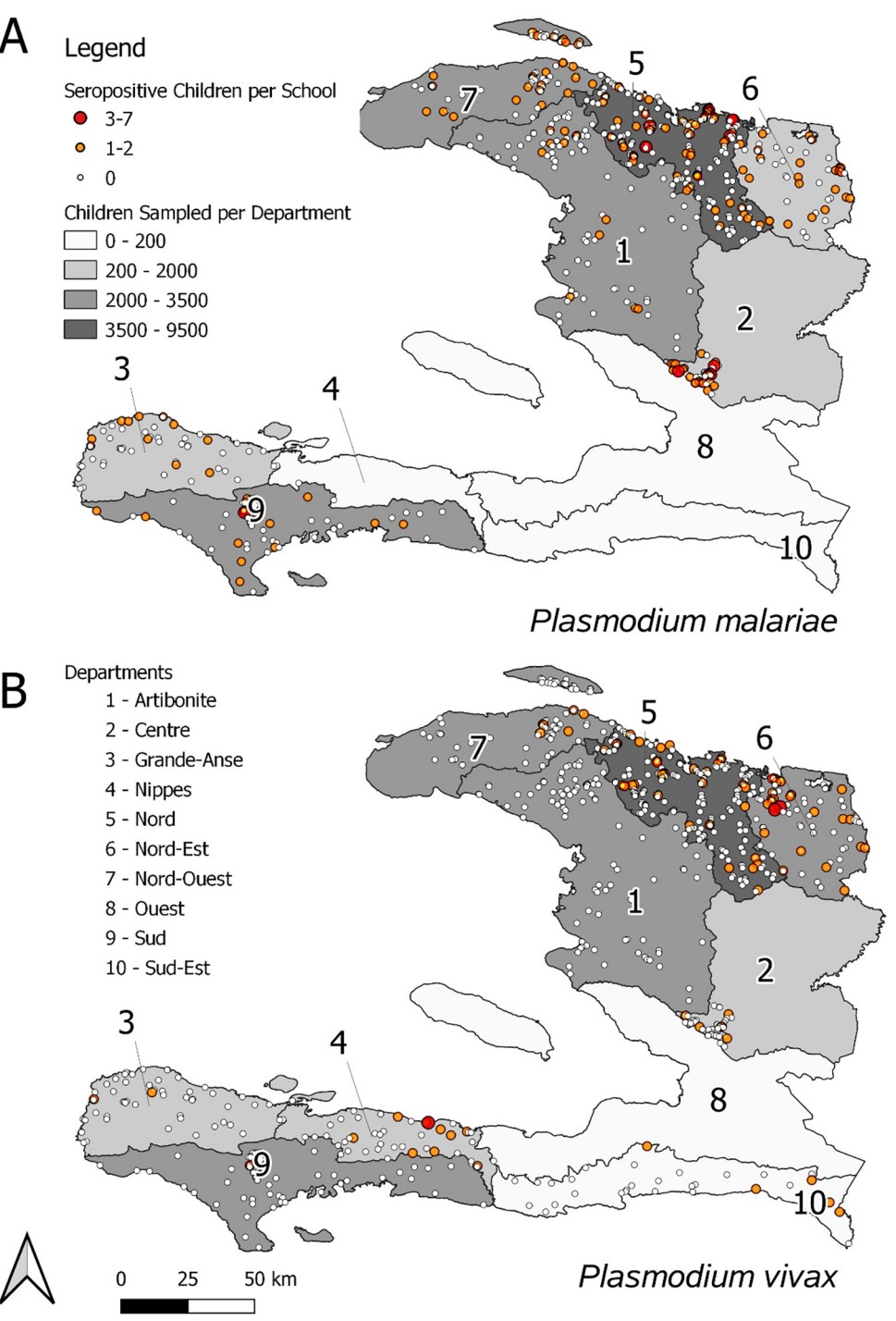

**Fig 2. Location of schools included in the TAS from 2014 to 2016 with malaria serology data collected.** Schools are represented by a dot where size and color change correspond to the count of children determined seropositive. White dot: no seropositive children; orange: 1–2; red: >3 seropositive children. Additionally, color intensity is provided for each Haitian department by total number of children with serology data enrolled within that department. Panel A shows seropositivity and enrollment for children with IgG data for PmMSP1$_{19}$, and panel B shows seropositivity and enrollment for children with IgG data for PvMSP1$_{19}$. Base map for administrative boundaries found at: https://www.geoboundaries.org/index.html#getdata.

weighted K-function of schools tested for *P. vivax* in northern Haiti also suggest a significant spatial aggregation of cases at all intervals between 2.5 km and 17.5 km with a peak at 12.5 km and again between 30 km and 50 km with a peak at 45 km (S1F Fig). Weighted K-functions for both species suggest spatial randomness at all distances in southern Haiti (S1D and S1H Fig). Although seropositive counts for *P. vivax* have no statistically significant aggregation at any interval, a very definitive arch can be noted between 5 and 50 km with a peak difference at 17.5 km (S1H Fig). This could suggest a borderline significant aggregation that has been underestimated by the spatial configuration of schools in the south (S1G Fig).

## Cluster analyses for P. malariae and P. vivax seropositivity

At a national level, spatial autocorrelation analysis using Moran's I resulted in values above zero for both *P. malariae* (I = 0.16, p<0.001) and *P. vivax* (I = 0.04, p = 0.13), suggesting statistically significant spatial autocorrelation on a national level for *P. malariae* but not for *P. vivax*. Moran's I for *P. malariae* suggests a significant autocorrelation of high seropositive counts in the north with an I-value of 0.153 (p<0.001) and no autocorrelation in the south with an I-value of -0.001 (p = 0.99)(S2 Fig). The Moran's I for *P. vivax* in the north does not suggest autocorrelation with an I-value of -0.016 (p = 0.80). *P. vivax* Moran's I in the south however does suggest autocorrelation but not to a significant level with an I-value of 0.062 (p = 0.08). Cluster location analysis of the entire country using Kulldorff's Bernoulii scan produced four hot-spots with a significant likelihood for seroprevalence: two for *P. malariae* and two for *P. vivax* (Table 3 and Fig 3). Two significant clusters were observed for *P. malariae* (Fig 3A): Cluster A (RR = 3.45, p<0.001) containing an ellipse with a minor radius of 9.1 km and major radius of 13.7 km, and Cluster B (RR = 5.38, p = 0.005) containing an ellipse with a minor radius of 2 km and major radius of 10 km. Two significant clusters were observed for *P. vivax* (Fig 3B): Cluster C (RR = 17.5, p<0.001) containing a circular hot-spot with a radius of 2.7 km and was located in the Nord-Est department, and Cluster D (RR = 5.48, p = 0.037) contained a circular hot-spot with a radius of 14.8 km was found in the southern department of Nippes. When Kulldorff's Poisson Spatial Scans were re-run in the northern and southern regions independently, clusters were verified (Figs 3 versus S2). The Kulldorff's Poisson Scan suggested an additional *P. vivax* cluster closer to the southeast border with the Dominican Republic (S2 Fig).

## Testing for active P. malariae and P. vivax infections through laboratory assays

In attempt to identify children with active *P. malariae* or *P. vivax* infections, blood samples with the highest IgG responses to the $PmMSP1_{19}$ and $PvMSP1_{19}$ antigens were selected to undergo laboratory assays for the presence of *Plasmodium* antigens or DNA. As shown in S3 Fig, the majority of all children's IgG signals to the PmMSP1 or $PvMSP1_{19}$ antigens were quite low (<10,000 MFI units on a scale of 0 to 32,000). In total, 27 samples were selected based on $PmMSP1_{19}$ antibody level, 24 samples selected based on PvMSP1 response, and 1 sample

**Table 3. Summary of hotspot clustering for *P. malariae* and *P. vivax* through Kulldorff's scan statistic among school children in Haiti.**

| Cluster | Species | No. Seropositive | Population | Schools | Major/Minor Radius (Km) | RR | P value |
|---------|---------|------------------|------------|---------|-------------------------|------|---------|
| A | P. malariae | 46 | 1182 | 49 | 9.1 / 13.7 | 3.45 | 0.001 |
| B | P. malariae | 18 | 255 | 8 | 2.0 / 10.0 | 5.83 | 0.005 |
| C | P. vivax | 12 | 170 | 5 | 2.7/ 2.7 | 17.05 | 0.001 |
| D | P. vivax | 12 | 521 | 13 | 14.8 / 14.8 | 5.48 | 0.037 |

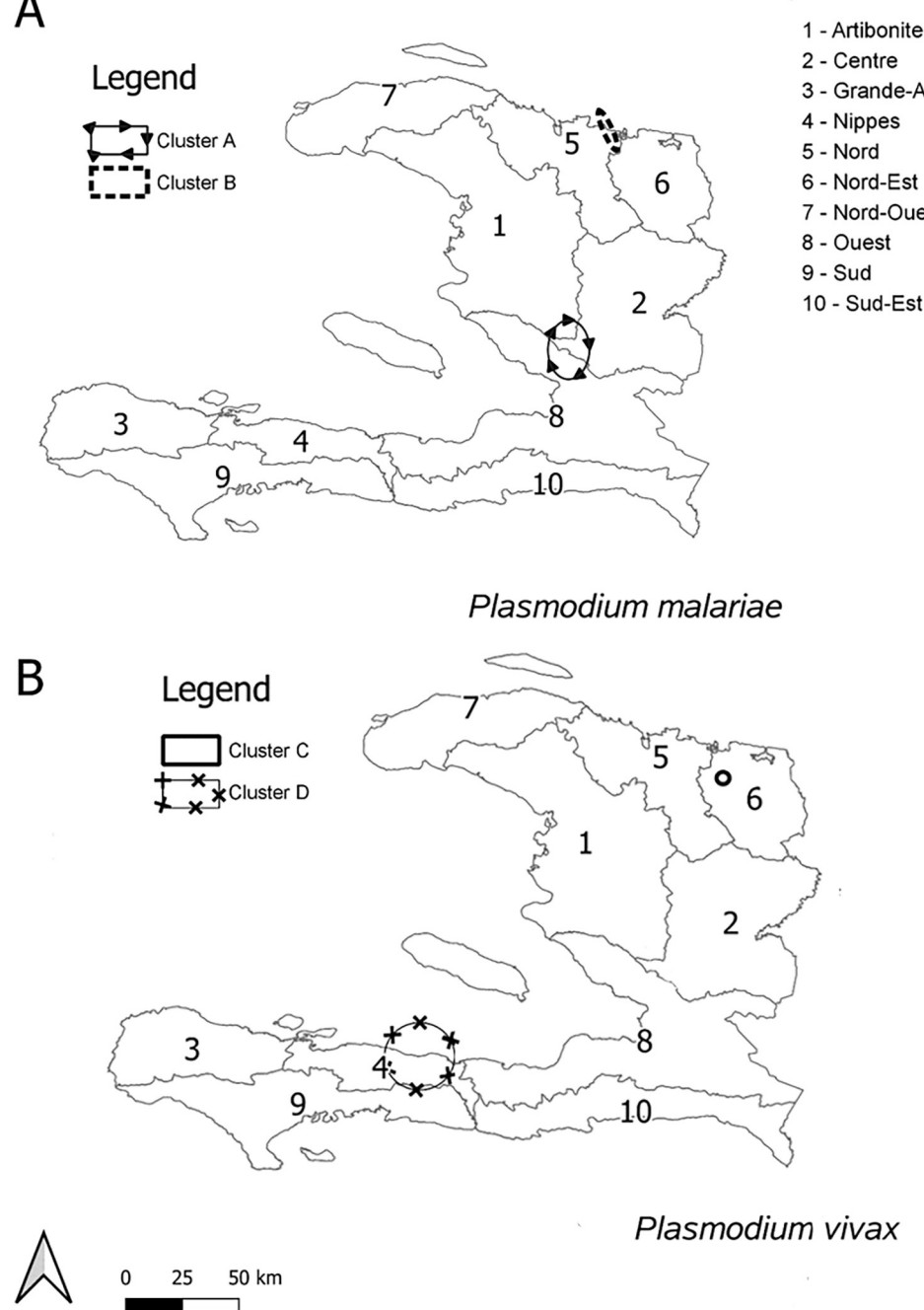

**Fig 3. Kulldorff's spatial scan statistic of seropositive children per school.** Maps displayed for *P. malariae* (A) and *P. vivax* (B) with ellipses denoting hotspot cluster boundaries for significant spatial aggregation of seropositive counts. Base map for administrative boundaries found at: https://www.geoboundaries.org/index.html#getdata.

selected due to high responses for both PmMSP1$_{19}$ and PvMSP1$_{19}$. For all 52 selected samples, neither *Plasmodium* antigen detection nor PET-PCR assays produced positive test results. As no antigen/DNA positives were found among these samples with the highest IgG responses, no further samples were selected for antigen and DNA testing.

## Discussion

Despite large-scale elimination efforts in the 1960s, malaria remains endemic in Haiti. The WHO-lead elimination program clearly identified three species of malaria in the population: *Plasmodium falciparum*, *P. malariae*, and *P. vivax* [13,14,32], but more recent malaria studies have largely overlooked the presence of non-*P. falciparum* species. The lymphatic filariasis (LF) TAS sampling methodology was developed to assess LF transmission [20,33], but school-based sampling designs are readily transferable to malaria exposure surveillance. The use of school children as a sentinel population with a biological peak determinacy of intensity and prevalence, is equally beneficial for malaria survey purposes [34–36]. With an extensive sample size of 21,719 children tested for antibodies against *P. malariae* and 24,559 children tested for *P. vivax*, this large seroepidemiological study also benefits from the very narrow age range of enrolled participants since high numbers of children of the same age (i.e. same number of years for possible exposure) were collected throughout the country. As IgG antibodies to *Plasmodium* antigens could have only been induced in this population from natural exposure, presence of IgG to these antigens in 6- or 7-year-old children would clearly indicate *P. malariae* or *P. vivax* exposure in these areas in the past 6 or 7 years. Recent studies have estimated that IgG against *Plasmodium* $MSP1_{19}$ antigens would have a duration of years to decades following natural exposure [9,11]. As the persons enrolled in this study were young, it is a reasonable assumption that presence of this long-lived IgG against the $MSP1_{19}$ antigens is a reliable indicator of lifetime *P. malariae* and/or *P. vivax* exposure.

For all Haitian children sampled, the IgG seroprevalence was very low for both species' $MSP1_{19}$ antigens: *P. malariae* (1.3%) and *P. vivax* (0.5%). The IgG response to the two $MSP1_{19}$ antigens appeared to be largely independent from each other (as well as independent from $PfMSP1_{19}$), and the total number of children seropositive to both $PmMSP1_{19}$ and $PvMSP1_{19}$ was small (n = 12, 0.06%). To the authors' knowledge, the most recent surveys to investigate the presence of *P. falciparum*, *P. malariae*, and *P. vivax* in Haiti was through the WHO elimination program with data from 1964 to 1967 [13]. During these four years, *P. falciparum* remained the dominant malaria species, accounting for approximately 98% of all microscopically-confirmed cases, whereas *P. malariae* accounted for ~2.5% and *P. vivax* ~0.5% of all case counts. It is not surprising that, even today, *P. falciparum* remains the dominant malaria in Haiti [12,37]. Though we present here data showing IgG concordance with antibodies against $PfMSP1_{19}$, a complete report of *P. falciparum* seroprevalence and spatial estimates will be forthcoming. From the serological data in our current study, it is interesting to note that *P. malariae* seroprevalence is over 2-fold higher than *P. vivax*–providing evidence that *P. malariae* may still be the second most prevalent malaria species in Haiti followed by *P. vivax*. Recently, more evidence has been generated that *P. vivax* can sustain transmission in largely-Duffy negative populations through alternate blood cell invasion routes [38–41]. This appears to be the situation in Haiti as well, with a largely Duffy negative population [42] showing a very low transmission rate for *P. vivax* [15,17].

When looking at seroprevalence by Haitian department, substantial differences were noted between IgG carriage against the *P. malariae* and *P. vivax* $MSP1_{19}$ antigens. However, even with significant differences among the department seroprevalence estimates, the absolute magnitude of IgG positivity for these antigens was still very low with the highest $PmMSP1_{19}$ seroprevalence at 3.8% (Centre department) and the highest $PvMSP1_{19}$ seroprevalence at 3.3% (Sud-Est department).

Analysis describing the distribution of the schools through un-weighted K-functions determined whether any clustering was simply a result of the spatial location of the sample sites. Ripley's K-function for spatial randomness determined that schools are spatially clustered at all intervals tested for north and south Haiti for both species. Therefore, any clustering of

seropositive children could potentially be dependent on school location and therefore be over-estimated due to the aggregation of school sample locations [29,30]. The suggested unweighted spatial distribution of schools calculated through this software was significantly clustered at all intervals for both species. Although the weighted K-function for *P. malariae* and *P. vivax* in north Haiti suggests borderline aggregation of cases across various intervals and Moran's I for *P. malariae* in the north suggest autocorrelation, all other autocorrelation measures clearly suggest that high seropositive counts do not neighbor other high counts–i.e. these areas are isolated from each other. If dividing the country into northern or southern sections or maintaining the area of the country as a whole, the overall concordance in defining areas of statistically significant seropositive clustering (Figs S2 versus S3) provides higher confidence in the finding of true geographical areas with higher *P. malariae* or *P. vivax* exposure. It is interesting to note that: 1) among the 4 clusters identified, none were overlapping between *P. malariae* and *P. vivax*, and 2) very few places in the country had a complete absence of children seropositive for either species. Practically, detection of hotspots of seropositivity for non-dominant malaria species could provide rationale for follow-up studies in these specific areas in attempt to obtain more granular spatial estimates for tangible programmatic purposes such as targeted mass-drug administration (MDA) in smaller populations. Additionally, health facilities within hotspot areas could be put on notice and given appropriate diagnostic tools to take into account non-dominant species when diagnosing a patient with 'malaria-like symptoms'.

A limitation to this current study is that children were enrolled at their respective schools, and not at primary residence where they would likely be spending the majority of their time, especially nights when the Anopheline mosquitoes are more active [43]. In this same manner, previous travel history for these children remains unknown and therefore malaria exposure (and IgG acquisition) could have occurred in a different area than at the area of the country where they were sampled. In addition, it is important to note that the spatial tests used serve as a descriptive measure of case distribution and can only determine general locations of high prevalence but not the magnitude of difference. The methods also do not suggest a high measure of granularity of malaria exposure.

As active infection with *P. vivax* or *P. malariae* is known to induce high levels of IgG, we selected the highest IgG responders to PmMSP1$_{19}$ and PvMSP1$_{19}$ in an attempt to identify current infections at time of sampling. Through qPCR and antigen detection assays, we were unable to identify any active malaria infections, so a limitation to this study is that no assertions could be made about *P. malariae* or *P. vivax* prevalence during the time of sampling between 2014–2016. Though providing an overall large sample size, many areas of Haiti did not have schools enrolled in the 2014–2016 TAS, so no assumptions could be made regarding malaria exposure in those areas not sampled. The sampling distribution of schools and low parasite prevalence of non-falciparum species in Haiti may also bias geospatial models as a large amount of empty space may exist between aggregate clusters of schools that were sampled [3,29,44,45].

In Haiti and other areas of the world where non-dominant malaria species are neglected in the country's surveillance and diagnostic efforts, serological data provides an objective measure to estimate population-level exposure to all endemic malaria species. In addition, as the design of diagnostic tests is intended to identify symptomatic and high parasite density infections, the use of IgG data can assist to approximate all forms of malaria exposure and infection, not just those in the population seeking treatment.

## Supporting information

**S1 Fig.** K-functions and weighted K-functions detailing spatial randomness of schools surveyed for IgG antibodies against P. malariae (A-D) and P. vivax (E-H) in Haiti. Spatial scan

area was divided into northern and southern regions in order to avoid empty space bias caused by lack of sampling in the Ouest department. For unweighted fields, confidence envelopes are generated by distributing points randomly in the study area and calculating k-values for 999 permutations. For each distance, the highest and lowest deviation from the expected K-value construct the envelope. The same methods are used for weighted fields with the exception that only the weighted values are randomly distributed to generate confidence envelopes; point locations remain fixed.
(TIF)

**S2 Fig.** Kulldorff's Spatial Scan for seropositive counts for *P. malariae* (top panel) and *P. vivax* (lower panel) as divided into the northern and southern sections of the country. Ellipses denote cluster borders, and shading of sub-communes indicate number of seropositive children enrolled during the TAS. Base map for administrative boundaries found at: https://www.geoboundaries.org/index.html#getdata.
(TIF)

**S3 Fig.** Selection of samples with high IgG assay signal to PmMSP119 (A) and PvMSP119 (B) for further laboratory tests. Samples providing assay signals in the grey circles were selected.
(TIF)

**S1 File. Raw data used for analysis.**
(XLSX)

## Acknowledgments

The authors would like to acknowledge the work of the Haitian field teams and the involvement of participants in this study. The findings and conclusions in this report are those of the authors and do not necessarily represent the official position of the CDC.

## Author Contributions

**Conceptualization:** Adan Oviedo, Eric Rogier.

**Data curation:** Camelia Herman, Eric Rogier.

**Formal analysis:** Adan Oviedo, Eric Rogier.

**Investigation:** Eric Rogier.

**Project administration:** Alaine Knipes, Caitlin M. Worrell, LeAnne M. Fox, Luccene Desir, Carl Fayette, Alain Javel, Franck Monestime, Kimberly E. Mace, Michelle A. Chang, Jean F. Lemoine, Kimberly Won, Venkatachalam Udhayakumar.

**Resources:** Kimberly Won, Venkatachalam Udhayakumar.

**Supervision:** Carl Fayette, Alain Javel, Franck Monestime, Jean F. Lemoine.

**Writing – original draft:** Adan Oviedo, Eric Rogier.

**Writing – review & editing:** Camelia Herman, Alaine Knipes, Caitlin M. Worrell, LeAnne M. Fox, Luccene Desir, Kimberly E. Mace, Michelle A. Chang, Kimberly Won, Venkatachalam Udhayakumar.

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
