## [Decision Letter · Decision Letter 0]

11 Jul 2021

Dear Dr. Rogier,

Thank you very much for submitting your manuscript "Spatial Cluster Analysis of Plasmodium vivax and P. malariae Exposure Among Haitian School Children Sampled Between 2014 And 2016" for consideration at PLOS Neglected Tropical Diseases. As with all papers reviewed by the journal, your manuscript was reviewed by members of the editorial board and by several independent reviewers. In light of the reviews (below this email), we would like to invite the resubmission of a significantly-revised version that takes into account the reviewers' and editors' comments. 

We cannot make any decision about publication until we have seen the revised manuscript and your response to the reviewers' comments. Your revised manuscript is also likely to be sent to reviewers for further evaluation.

Sincerely,

Daniel M Parker

Associate Editor

Amanda Ross

Deputy Editor

Reviewer's Responses to Questions

**Key Review Criteria Required for Acceptance?**

**Methods**

-Are the objectives of the study clearly articulated with a clear testable hypothesis stated?

-Is the study design appropriate to address the stated objectives?

-Is the population clearly described and appropriate for the hypothesis being tested?

-Is the sample size sufficient to ensure adequate power to address the hypothesis being tested?

-Were correct statistical analysis used to support conclusions?

-Are there concerns about ethical or regulatory requirements being met?

Reviewer #1: See all comments below

Reviewer #2: - Provide a clearer explanation on the differences between each of the three spatial/cluster analyses used. Each of the sections in the results talk about “clustering” but its hard to tell how the interpretations between the three methods fit together. 

- Provide details in the statistical analysis section on the use of Bayesian logistic model, and it is not clear in the results as to why this was performed and why the OR estimates were inconsistent for NDVI and LST. This finding seems very strange for univariate multilevel logistic models – what were the priors, how influential? Also for the multilevel logistic model more details should be provided on the random effects – e.g. for school clustering.

Reviewer #3: The authors clearly stated their objective to establish seroprevalence of malaria species in Haiti. They used a young age group collected at schools to do this in order to look at recent infections and were able to establish appropriately large sample sizes (21,719 students from 680 schools). They used appropriate statistics to test for the degree of spatial aggregation of the schools, and autocorrelation of seropositivity counts, followed by Kulldorff’s Bernoulii scan to identify significant hotspots.

**Results**

-Does the analysis presented match the analysis plan?

-Are the results clearly and completely presented?

-Are the figures (Tables, Images) of sufficient quality for clarity?

Reviewer #1: See all comments below

Reviewer #2: - Figure 1: need to have higher resolution on these figures.

- Table 1 – 2: It should be made clearer in Tables 1 and 2 that the OR presented for age, elevation, NDVI and LDT are for these as continuous covariates (also did the authors investigate if the association between these continuous covariates and the log odds of seropositivity was linear, as no details are provided in the statistical analysis section), and the units should be given in the variable column for age. Also the reference group for gender should be specified. ‘Variable’ is not spelt correctly in Table. 

- Line 343 – 347: the 95% CI should be provided in brackets next to the OR estimate, and the exact p-value instead of a categorised p-value. (Should be “<” instead of “>” on line 343 – 344 if you decide to keep these p-values in text). 

- Line 363: change “detail” to “correspond to”.

- Figure 2: consider giving one common legend for this figure rather than splitting between the two plots.

- Figure S2: explain the acronyms shown in the figures (L(d), Low Conf. Env., High Conf. Env.) in the figure legend.

- Line 389 - 414: give actual p-value rather than categorising as >0.05 and <0.01. Replace by 95% CI if this is available. 

- Table 3: include information on method used in the legend, for example: “by Kulldorff’s scan statistics.” Please also provide confidence intervals for the RR estimates in the table and in text if available. 

- Line 426: can remove “as described in Methods”. 

- Line 426: remove the ’ at the end of children’s.

Reviewer #3: The results establish the geographical distribution of cases of various malaria species in Haiti as the authors intended. Drawing figure 3 as a superimposition over cases, as was done in S3, may help readers see the pattern more effectively.

**Conclusions**

-Are the conclusions supported by the data presented?

-Are the limitations of analysis clearly described?

-Do the authors discuss how these data can be helpful to advance our understanding of the topic under study?

-Is public health relevance addressed?

Reviewer #1: See all comments below

Reviewer #2: - The discussion could be made stronger by having a summary of the key findings, which address the research questions detailed in the introduction, earlier in the discussion section (at least in the first paragraph). 

- Line 462: consider removing the word “vanishingly”, small is enough. 

- Line 512 – 513: “autocorrelation measures clearly suggest that high seropositive counts do not neighbor other high counts” – but in the results section autocorrelation results showed clustering of high P. malariae seropositivity in the north? 

- Include in the discussion section recommendations on how the detection of hotspots using Kulldorff’s scan statistic (Table 3) could be used – do the authors recommend anything based on these hotspot results?

Reviewer #3: Conclusions are generally well supported by the data presented and limitations are mostly well addressed. One exception would be the exclusion of P. ovale within the study is not mentioned when discussing relative species prevalence.

**Editorial and Data Presentation Modifications?**

Reviewer #1: See all comments below

Reviewer #2: - Line 1-3: consider including something about seroprevalence or serological data in the title.

- Line 28 – 29: include study aims at the end of the background section of the abstract. 

- Line 33: can remove the word “later” for simplicity. 

- Line 56 – 57: consider rewording this sentence to: “P. falciparum is the dominant malaria species worldwide and is often the primary, or only, focus of malaria surveys.”

- Line 58 and 62 – change “populace” to “population”. 

- Line 70 – 71: consider rewording to “Identification of Plasmodium reservoirs in low transmission settings rely on conventional diagnostics (microscopy and RDT) which underestimate the true incidence and prevalence.”

- Line 72 – 73: antibody-based assays are a type of diagnostic. We would suggest rewording “When compared to diagnostic tests” to “When compared to microscopy and RDTs”.

- Line 73: suggest removing “more powerful indicator for detecting active infection…” in place of “more powerful method for detection of active infection…”.

- Line 74 – 75: consider simplifying the last part of this sentence to “expanding the time window for diagnosis.” 

- Line 88: consider changing “reduction efforts” to “elimination efforts”. 

- Line 100 – 104: an overview of what is done in the study is given but should also have clear aims stated.

- Line 100: remove the word “occurring”.

Reviewer #3: In figure S3, in the top panel, I suggest the P-value be written using scientific notation so that it is not P < 0.000

Line 132: extra parenthesis at the end of the sentence

Line 142: missing space

Line 379: S1F figure should likely be S2F

Line 399: Combine table & figure references to be more consistent with previous reference formats

Line 442-444: Sentence lacks appropriate subject

**Summary and General Comments**

Reviewer #1: Oviedo and colleagues describe results from a large-scale sero-survey of >20,000 school children in Haiti, with a particular focus on P. malariae and P. vivax. They identified low-levels of sero-positivity against both P. malariae and P. vivax MSP1-19, with evidence of some (independent) spatial clusters. In a subset of samples tested, they found no evidence of active Plasmodium infections. The article is well written, logical and with sound data analysis. I have only minor comments that I as a reader would find helpful for the authors to address.

1. Cross-reactivity between Pf, Pm, Pv MSP1-19

The authors seem confident that there would be minimal cross-reactive antibody responses against the various MSP1-19 proteins tested from the three species, based on past literature and their results in Figure 1. 

a. I suggest in the methods that the authors do detail the sequence identity between the Pf-Pv Pf-Pm and Pv-Pm MSP1-19 protein sequences (I believe it is moderate at least for Pv-Pm around 56%).

b. In the results description for Figure 1, it would be helpful to quantify how many children had a positive signal to more than one MSP1-19 antigen out of the total children positive with any positive signal. This is detailed around lines 318/319 for Pm and Pv but doesn’t include Pf, and clearly from Figure 1 there are quite a few positive antibody signals for Pf.

2. Results for Pf MSP1-19

I appreciate that the authors have focused their results on Pv and Pm given the prevalence/transmission of these species in recent years in Haiti remains unknown. However they did measure antibody against PfMSP1-19 so I think it is worth at least documenting/quantifying the proportion of the children seropositive to Pf-MSP1-19 even if the subsequent detailed analysis is only on Pm and Pv. This could be added to the text description for Figure 1, and added into Line 459 in the discussion.

3. Remaining minor points

a. Line 75 seems to be an aberrant comma

b. Line 81 define MSP1 for first use

c. Line 142 missing space between C and until

d. Line 164 text says City, STATE – needs adding in

e. Lines 155, 173, 204 please provide more details. What is the positive control? How many negative controls were run to generate the seropositivity cut-off?

f. Would Figure 1B be clearer on a log scale like Figure 1A?

g. Line could be helpful to give more context – latest % prevalence estimate? Or # cases per year?

h. Line 456 would antibody longevity be expected to be similar between adults and children?

i. In the discussion, the authors could comment on their use of binary seropositivty data, would added value be gained from using the actual antibody magnitude?

j. For the discussion, can the authors comment on whether there are any geographic or human features that associate with the clusters? Like village boundaries, or rivers/mountains/roads etc? And perhaps why Cluster D is so large (is this an effect of less schools being tested in that region)?

k. Do the authors think it is worthwhile doing their PCR assays/antigen assays on the whole sample set in the future to search for active Pm and Pv infections or would that be waste of time/resources?

I selected "no" for is all the underlying data available as it was not included as a Supplement so I personally could not access it. The corresponding author does say it is available upon request so the Editor can determine whether this is appropriate or not.

Reviewer #2: The manuscript “Spatial cluster analysis of Plasmodium vivax and Plasmodium malariae exposure among Haitian school children sampled between 2014 and 2016” presents an interesting analysis of a very important issue. As malaria declines it is essential that we understand the changing dynamics of transmission. This study will provide useful information to those working in malaria elimination in Haiti and those who are interested in how serological data can supplement existing diagnostic methods. 

The comments provided here recommend further improvements to the manuscript.

Reviewer #3: The study by Oviedo et al provides a serosurvey of P. malariae, P. vivax and P. falciparum in Haiti between 2014 and 2016. The study focuses on young children who have just a few years of potential exposure so the data presented is valuable in its ability to inform scientists on the degree to which these species have been endemic in Haiti over the last few years. The use of spatial clustering methods established areas where each species is likely to be most endemic and may be useful in elimination efforts. The methodology seems appropriate for the conclusions. While there are some limitations that result from sampling bias, the authors address this multiple times in their results and discussion. I recommend this manuscript be accepted, though I have some minor recommendations: 

Sources 5 and 23 both appear to reveal plausibility for detection of P. ovale, and the authors mention that the last surveys in this area were in the 60s. Further, they assert that results may indicate P. malariae is the second most common on lines 470-472 without having data for P. ovale available. Authors should discuss why they chose to not include P. ovale and to consider this in their assertions. 

Authors should comment on why they chose not to adjust for collection or population numbers.

I also noted the following minor writing errors: 

Line 132: extra parenthesis at the end of the sentence

Line 142: missing space

Line 379: S1F figure should likely be S2F

Line 399: Combine table & figure references to be more consistent with previous reference formats

Line 442-444: Sentence lacks appropriate subject

Figure S3, in the top panel, I suggest the P-value be written using scientific notation so that it is not P < 0.000

Deputy editor: 

The statistical analysis appears a little weak. The methods chosen are not well justified, and there is a heavy reliance on scan statistics which while simple and easy to run can only say whether there is a significantly higher prevalence in an area but not how much higher (which would be useful for control activities). In figure 3, the hotspots for P vivax are very large, but the prevalence is unlikely to be uniformly high across this area, there are relatively few cases and it does not correspond well visually to Fig 2. (A prevalence map may be an option, although would require more sophisticated statistical methods). I suggest reducing the emphasis on the cluster analysis, and removing the section on spatial randomness since malaria is well known to be heterogenous on many scales. The size of the clusters does not seem to be justified - ideally it would bear relevance to control activities or whatever the information is used for. In Fig S3, the elipses cover areas with no data and so the shape seems rather inflexible. Whether you choose scan statistics or other analysis, the methods should be justified and limitations mentioned in the discussion.

For the univariate logistic regression models, clustering within schools does not look as if it has been taken into account, and the choice of univariate is neither obvious nor justified. A statistician would be able to advise on this. 

PLOS authors have the option to publish the peer review history of their article (what does this mean?). If published, this will include your full peer review and any attached files.

Reviewer #1: No

Reviewer #2: No

Reviewer #3: No
---

## [Decision Letter · Decision Letter 1]

19 Oct 2021

Dear Dr. Rogier,

Thank you very much for submitting your manuscript "Spatial Cluster Analysis of Plasmodium vivax and P. malariae Exposure Using Serological Data Among Haitian School Children Sampled Between 2014 and 2016" for consideration at PLOS Neglected Tropical Diseases. As with all papers reviewed by the journal, your manuscript was reviewed by members of the editorial board and by several independent reviewers. The reviewers appreciated the attention to an important topic. Based on the reviews, we are likely to accept this manuscript for publication, providing that you modify the manuscript according to the review recommendations. 

Sincerely,

Daniel M Parker

Associate Editor

Amanda Ross

Deputy Editor

Reviewer's Responses to Questions

**Key Review Criteria Required for Acceptance?**

**Methods**

-Are the objectives of the study clearly articulated with a clear testable hypothesis stated?

-Is the study design appropriate to address the stated objectives?

-Is the population clearly described and appropriate for the hypothesis being tested?

-Is the sample size sufficient to ensure adequate power to address the hypothesis being tested?

-Were correct statistical analysis used to support conclusions?

-Are there concerns about ethical or regulatory requirements being met?

Reviewer #1: The authors addressed my comments re some additional clarifications in the serological methods appropriately.

Reviewer #2: (No Response)

Reviewer #3: (No Response)

**Results**

-Does the analysis presented match the analysis plan?

-Are the results clearly and completely presented?

-Are the figures (Tables, Images) of sufficient quality for clarity?

Reviewer #1: The authors considered my suggestions and ultimately disagreed, which they appropriately justified.

Reviewer #2: (No Response)

Reviewer #3: (No Response)

**Conclusions**

-Are the conclusions supported by the data presented?

-Are the limitations of analysis clearly described?

-Do the authors discuss how these data can be helpful to advance our understanding of the topic under study?

-Is public health relevance addressed?

Reviewer #1: The authors have better acknowledged the limitations of their study following a suggestion by the editor.

Reviewer #2: (No Response)

Reviewer #3: (No Response)

**Editorial and Data Presentation Modifications?**

Reviewer #1: In all Tables, needs to have the Plasmodium species in italics.

In Table 1 and Table 2, the "a" after schools could be in superscript then it would look less like a typographical error.

For clarity, I would suggest noting in the results or discussion that analyses and presentation of the Pf data will be in an upcoming publication.

Reviewer #2: Line 59: change “populace” to “population” – make sure “population” used throughout text for consistency

Line 66: change “test for antibodies” to “testing for antibodies” or “tests for antibodies”

Line 67: change “burden to” to “burden of”

Line 72: change “rely” to “relies”

Line 88: change “of years” to “for years”

Line 105: change “will only identify” to “only identify”

Line 119: change “helminth” to “helminths” 

Line 295 – 297: would be helpful to have an estimate of the area covered by the north and south regions, and the region as a whole (in km2 for example) to give context into how large each of these sub-areas are compared to the whole region. 

Line 336 – 342: when you specify the departments with more than 100 children sampled you should also state species to make it clear which you are talking about. For example: “For departments with more than one-hundred children sampled, P. vivax seroprevalence was highest in Sud-Est (3.3%)…” 

Line 344: would be helpful to have a range alongside the average age of positive children. 

Line 348: use a symbol like a cross or star instead of a, or make “a” a superscript to make it clear it is a comment not a misplaced letter

Line 388: remove “the” from “in the northern”

Line 395: make sure “k” in K-function is capitalized 

Line 395: could be good to remind the reader here how the K-functions are weighted 

Line 431: make sure malaria spp is italicized throughout text (also unitalicized on line 528)

Line 460: remove “which” 

Line 521: remove “not”

Reviewer #3: In figure 3, I maintain that adjusting for sampling would be useful. the number of seropositive students per school isn't very informative if sampling isn't entirely uniform between schools and therefore I think the color coding of schools by seropositive students per school could be misleading. I believe this to be a rather minor concern that doesn't threaten overall conclusions.

**Summary and General Comments**

Reviewer #1: (No Response)

Reviewer #2: The authors have responded well to the first round of comments and suggestions which have improved the manuscript. The comments provided here recommend further improvements to the manuscript. Line numbers provided refer to the location in the tracked changes document.

Reviewer #3: (No Response)

PLOS authors have the option to publish the peer review history of their article (what does this mean?). If published, this will include your full peer review and any attached files.

Reviewer #1: No

Reviewer #2: No

Reviewer #3: No

Deputy editor: 

The authors have improved the paper.

I remained concerned about the implausibly large hotspot – it does not tally with the data in Figure 2. I think this is likely to be just a matter of changing the maximum size of the scan window to something justifiable. (It does not need to be justifiable on the grounds of interventions if they are not to be used, but should be justifiable for some reason other than that it was the default software setting).

I agree with Reviewer 2 (L404-430) who requested actual p-values rather than p<0.05 or p<0.01. Statistics has moved away from these cut-offs now and textbooks recommend using the exact p-value since they contain more information and also give a more accurate impression of the test that was carried out. Please use the exact p-values.

The sentence added at the end of the Introduction about the rationale for the analysis does not seem accurate. An ‘analysis’ does not aim to provide a rationale for something, it only answers a specific question. Do you mean the aim of the ‘study’ is to provide a rationale for something else? It would make more sense if the aim of the study would be to summarize the spatial clusters of Pf and Pm in Haiti, and to demonstrate the use of IgG. Otherwise it does rather sound as if the aim of research is to justify research. 

Figure Files:

Data Requirements:

Reproducibility:

References

---

## [Editor Report · Decision Letter 2]

30 Nov 2021

Dear Dr. Rogier,

Thank you very much for submitting your manuscript "Spatial Cluster Analysis of Plasmodium vivax and P. malariae Exposure Using Serological Data Among Haitian School Children Sampled Between 2014 and 2016" for consideration at PLOS Neglected Tropical Diseases. As with all papers reviewed by the journal, your manuscript was reviewed by members of the editorial board and by several independent reviewers. The reviewers appreciated the attention to an important topic. Based on the reviews, we are likely to accept this manuscript for publication, providing that you modify the manuscript according to the review recommendations. 

The authors have adequately addressed all reviewer critiques, with one exception. Please address the following issue:

Line 269 "The north region included the departments Nord, Nord-Est, Nord-Ouest, Artibonite, and Centre. The southern region included the departments, Grand'Anse, Sud, Nippes, and Sud-Est."

Please give total land areas for these subregions. As you state in the previous rebuttal, you do already provide the total landmass of Haiti. However, it is useful for the reader to understand the two subareas that are analyzed, especially since one part of the nation is excluded (Oueste). This should be easy to calculate/estimate with the GIS software used. Without this it would be easy for a reader to think that the entire nation was included in this analysis. Also consider a very brief statement about why Oueste is excluded.

Sincerely,

Daniel M Parker

Associate Editor

Amanda Ross

Deputy Editor

The authors have adequately addressed all reviewer critiques, with one exception. Please address the following issue:

Line 269 "The north region included the departments Nord, Nord-Est, Nord-Ouest, Artibonite, and Centre. The southern region included the departments, Grand'Anse, Sud, Nippes, and Sud-Est."

Please give total land areas for these subregions. As you state in the previous rebuttal, you do already provide the total landmass of Haiti. However, it is useful for the reader to understand the two subareas that are analyzed, especially since one part of the nation is excluded (Oueste). This should be easy to calculate/estimate with the GIS software used. Without this it would be easy for a reader to think that the entire nation was included in this analysis. Also consider a very brief statement about why Oueste is excluded.

Figure Files:

Data Requirements:

Reproducibility:

References

---

## [Editor Report · Decision Letter 3]

3 Dec 2021

Dear Dr. Rogier,

We are pleased to inform you that your manuscript 'Spatial Cluster Analysis of Plasmodium vivax and P. malariae Exposure Using Serological Data Among Haitian School Children Sampled Between 2014 and 2016' has been provisionally accepted for publication in PLOS Neglected Tropical Diseases.

Best regards,

Daniel M Parker

Associate Editor

Amanda Ross

Deputy Editor

---

## [Editor Report · Acceptance letter]

24 Dec 2021

Dear Dr. Rogier,

We are delighted to inform you that your manuscript, "Spatial Cluster Analysis of Plasmodium vivax and P. malariae Exposure Using Serological Data Among Haitian School Children Sampled Between 2014 and 2016," has been formally accepted for publication in PLOS Neglected Tropical Diseases.

Best regards,

Shaden Kamhawi

co-Editor-in-Chief

Paul Brindley

co-Editor-in-Chief
